# Nutritional, Metabolic, and Inflammatory Alterations in Children with Chylous Effusion in the Postoperative Period of Cardiac Surgery: A Descriptive Cohort

**DOI:** 10.3390/nu16223845

**Published:** 2024-11-10

**Authors:** Sidney V. Silva, Tais D. R. Hortencio, Lidiane O. S. Teles, Alexandre L. Esteves, Roberto J. N. Nogueira

**Affiliations:** 1Department of Pediatrics, State University of Campinas (UNICAMP), Campinas 13083-970, Brazil; 2Medicine Department, São Leopoldo Mandic Faculty, Campinas 13045-755, Brazil; tais.hortencio@slmandic.edu.br; 3Multidisciplinary Nutritional Therapy Team, State University of Campinas (UNICAMP), Campinas 13083-970, Brazil; lidi_enf@hotmail.com (L.O.S.T.); alexandreeslima@gmail.com (A.L.E.); 4Medical Clinical Department, State University of Campinas (UNICAMP), Campinas 13083-970, Brazil; nutrigene@uol.com.br

**Keywords:** chylous effusion, cardiac surgery, chylothorax, children

## Abstract

Objective: The occurrence of chylous effusion in children undergoing cardiac surgery is progressively increasing due to technical advances that have led to a rise in the number of surgeries. In this context, the objective was to describe the clinical profile of a cohort of patients at the time of chylous effusion diagnosis. Methods: A retrospective cohort analysis was conducted between January 2011 and July 2023, involving 23 patients, aged 0 to 18 years, treated at a quaternary university hospital in southeastern Brazil. Data were obtained from the follow-up records of the Multidisciplinary Nutritional Therapy Team (MNTT) for patients who received nutritional support after developing chylous effusion in the postoperative period of cardiac surgery. Results: The younger population predominated (median age of 6 months), with a high prevalence of malnutrition (60.9%). At the time of chylous effusion diagnosis, 83% had lymphopenia, and 74% had hypoalbuminemia. The longer the time elapsed after surgery for the onset of chylous effusion, the lower the HDL cholesterol, the lower the albumin levels, the greater the surgical complexity, the younger the patient, and the lower their weight. Hypocalcemia occurred in nearly half of the sample and hypophosphatemia in 26% of the analyzed cases. Conclusions: There was a notable presence of lymphopenia, hypoalbuminemia, and low HDL cholesterol, as well as a high incidence of mineral imbalances, particularly hypocalcemia and hypophosphatemia, which, if untreated, may lead to unfavorable outcomes. Therefore, clinical and laboratory monitoring of children in the postoperative period of cardiac surgery is important and can aid in the early diagnosis of chylous effusion and, consequently, in the timely initiation of treatment.

## 1. Introduction

Lymphatic and chyle circulation disorders in pediatric patients are present in various clinical situations [1]. The causes are generally classified into three groups: those that lead to obstructive impairment of lymphatic circulation, such as expansive lesions; diseases that increase venous system pressure and hinder lymphatic drainage (e.g., venous thrombosis, tuberculosis); and direct or indirect traumatic injuries to lymphatic vessels, often resulting from surgeries, such as cardiac surgery [1].

With technological advancements in intensive care and surgical techniques, the number of patients eligible for correction of congenital heart malformations has increased, leading to a rise in the number of cases of chylous effusion directly or indirectly related to surgical procedures, as well pointed out by Mery et al., describing an increase in incidence from 2 to 3.7% between 2004 and 2011 [2,3]. The incidence of chylous effusion in children during the postoperative period for cardiac surgery, however, varies widely between 0.25% and 8.8% [1,2,4].

It is known that the occurrence of chylous effusion in the pleural cavity (chylothorax), abdominal cavity (chylous ascites), or pericardial cavity (chylopericardium) will result in longer hospital stays, increased costs, prolonged intensive care unit (ICU) stays, and higher morbidity and mortality [1,2,4]. Specialized nutritional intervention and, in some cases, additional surgical intervention will be required [2]. Due to the presence of chyle in body cavities, there will be a loss of plasma lipoproteins, resulting in a reduction in circulating proteins such as albumin and immunoglobulins, as well as a decrease in circulating lymphocytes [5,6,7].

Additionally, chylous fistula will cause the loss of fluids and minerals, leading to imbalances in electrolytes and acid-base levels, with potentially lethal consequences if not promptly corrected [5,6,7]. Ideally, we should prevent these complications, and the most effective way to do it is through proper nutritional and metabolic support. However, despite some management protocols for these cases, there is no consensus [6,7,8].

Thus, studying the clinical, nutritional, and metabolic variations related to chylous effusion is a crucial first step in developing protocols to help manage this serious condition. The objective of this study was to describe the clinical, nutritional, inflammatory, and metabolic characteristics of a cohort of children with chylous effusion in the postoperative period of cardiac surgery, analyzing possible correlations between clinical and laboratory findings at the time of effusion diagnosis.

## 2. Methods

This is a retrospective cohort study based on data collected from records archived by the Multidisciplinary Nutritional Therapy Team (MNTT) at the University of Campinas (UNICAMP) Clinical Hospital. The records are standardized and specifically designed to ensure uniformity of patient data. The study period ranged from January 2011 to July 2023.

All patients’ records aged from birth to under 18 years old who underwent surgical correction of congenital heart disease and, during the clinical recovery period until hospital discharge, developed some form of chylous effusion and subsequently received nutritional support from the MNTT were included. Patients diagnosed with pleural, abdominal, or pericardial cavity effusion, whether or not it appeared macroscopically milky, and with a triglyceride level in the effusion fluid equal to or greater than 110 mg/dL [1,9], were considered eligible.

Data extracted and analyzed included sex and age distribution, weight at the time of chylous effusion diagnosis, and nutritional status classification assessed by the World Health Organization (WHO) weight-for-age indicator, considering the patient’s chronological age [10].

The type of cardiac surgery performed was grouped according to the RACHS-1 classification [11]. This classification considers the complexity of the procedure and postoperative complications, categorizing surgeries into six groups based on expected mortality rates (Appendix A). The time between the surgical procedure and the onset of chylous effusion was recorded, along with patient outcomes.

Serum levels of lymphocytes, albumin, sodium, potassium, phosphorus, calcium, and magnesium at the time of chylous effusion diagnosis were also extracted and analyzed. The methods used by the Clinical Pathology Laboratory at the Clinical Hospital were as follows: lymphocytes: an automated method using fluorescent flow cytometry, impedance, hydrodynamic focusing, and sodium lauryl sulfate; sodium, potassium, and ionized calcium: amperometry and potentiometry; magnesium: colorimetric/xylidyl blue; phosphorus: UV phosphomolybdate; albumin: colorimetric/bromocresol green; creatinine: enzymatic colorimetric; HDL: enzymatic colorimetric; CRP: turbidimetry.

The assessment of adequacy (and classification of detected disorders) was based on the normal reference values of the Clinical Pathology Laboratory at the Clinical Hospital according to the patients’ age, where applicable, as follows: sodium for hyponatremia—1 month to 1 year: <129 mmol/L, >1 year: <135 mmol/L; potassium for hypokalemia—1 month to 1 year: <3.6 mmol/L, >1 year: <3.1 mmol/L; calcium for hypocalcemia—<1.15 mmol/L; phosphorus for hypophosphatemia—<4 mg/dL; magnesium for hypomagnesemia—up to 6 years: <1.2 mEq/L, 6 to 12 years: <1.2 mEq/L; lymphocytes for lymphopenia—under 2 years: <3000 × 10^6^/L, over 2 years: <1500 × 10^6^/L; and albumin for hypoalbuminemia—<3.5 g/dL.

This study was reviewed and approved by the UNICAMP Research Ethics Committee under registration CAAE 59914922.1.0000.5404 on November 2022. A waiver for obtaining informed consent was requested and granted, as no patient identification was involved, and they were no longer in contact with the health service at the time of this study, making them unreachable.

Descriptive analysis was used to characterize the study variables, based on absolute and relative frequencies for categorical variables, and on mean, median, standard deviation, and minimum and maximum values for continuous variables. The Kolmogorov–Smirnov test was applied to identify the normality of variable scores. Non-parametric inferential statistics were used, as the data did not follow a normal distribution. The correlation between the time of chylous effusion onset and outcome variables was assessed using Spearman’s test. The significance level was set at 5%. The analysis was performed with SPSS 16 (IBM, Chicago, IL, USA).

## 3. Results

Out of the 23 patients, 13/23 (57%) were male. Hypoalbuminemia was observed in 17/23 (73.9%) of the patients. Malnutrition was observed in 14/23 (60.9%) patients. Nineteen patients were discharged after completing treatment 19/23 (82.6%), two were transferred during treatment to their originating facility, and two deaths occurred (8.7%). According to the RACHS-1 classification, this sample included five patients in group 1, three in group 2, eleven in group 3, and four in group 4 (Table 1). The prevalence of hyponatremia, hypokalemia, hypomagnesemia, hypophosphatemia, lymphopenia, and hypoalbuminemia were described in Table 1.

The time elapsed between surgery and the diagnosis of chylous effusion, expressed in days, with a median (p25, p75) of 5 (3, 15) days.

The absolute laboratory values of patients at the time of chylous effusion diagnosis are presented in Table 2.

The triglyceride levels in the chylous effusion had a median of 423 mg/dL. The median age of the patients was 4.0 years, and they had a median weight of 4.8 kg.

We sought to correlate the initial laboratory findings of these patients with the time elapsed post-surgery for the appearance of chylous effusion. Albumin, RACHS-1, age, and weight were associated (*p* < 0.05) with time of chylous effusion onset (Table 3) using Spearman’s correlation test. A negative Rho (Spearman’s rank correlation coefficient) means that we found an inversely proportional correlation between these variables.

## 4. Discussion

In our sample, hypocalcemia and hypophosphatemia were much more prevalent than described for usual pediatric hospitalized patients. The main reason for this fact is the characteristic of the sample, composed of critically ill and inflamed patients, with a marked release of procalcitonin [12]. This fact itself may contribute to hypocalcemia [13]. Additionally, patients on fasting usually receive “maintenance fluids” containing glucose, sodium, and potassium, without the addition of phosphorus and calcium. The absence of these minerals in the “maintenance fluids” associated with the shift of phosphorus to intracellular compartment after glycose infusion and insulin release may contribute to the hypophosphatemia [14].

At the time of chylous effusion diagnosis, lymphopenia prevalence was 83% and hypoalbuminemia was 74%. These changes can be explained by two mechanisms: the loss of lymphocytes and plasma proteins into the chylous effusion [5,6,7,15] and the hypercatabolic state inherent to the postoperative period [8,15,16]. This latter mechanism can be confirmed by the frequent presence of low serum HDL cholesterol levels. Decreased HDL is a recognized and prevalent inflammatory marker [15]. These data show us a hypercatabolic status that could contribute to the massive release of inflammatory hormones with a consequent hyperglycemia and insulin release that may cause shift of intracellular minerals, like phosphorus, into the cells.

In fact, this finding of our study is particularly relevant because in other situations, the most common mineral disturbances are sodium and potassium disorders [17]. Furthermore, sodium and potassium disorders are more closely monitored and promptly corrected.

Therefore, monitoring all these minerals, including calcium and phosphorus, in patients with a high likelihood, suspicion, or diagnosis of chylous effusion is essential, as such metabolic disturbances worsen clinical outcomes, whether in surgical patients or not [17,18,19].

Considering that this study aimed to analyze patients who underwent cardiac surgery, the age distribution observed, with a predominance of young infants (median age of 6 months), is easy to understand. Indeed, in cases of congenital heart disease, there is a significant portion of cases that require early surgical correction [8]. Another observation was the median weight of 4.8 kg, with a clear predominance of malnutrition in the sample (60.9%). This can be explained by the hypercatabolic state and the frequent need to restrict daily fluid intake in these patients. The nutritional status of critically ill patients, whether pediatric or not, is crucial to their clinical progression. Numerous studies show that malnutrition, which can be assessed and classified in various ways, is associated with multiple complications, metabolic disorders, and consequently, worsened clinical outcomes, whether in surgical or non-surgical patients [2,15,16,20,21].

Malnutrition was notably detected in 61% of the sample in this study, much higher than the cohort described by Buckley et al. [16], which reported approximately 25% of patients with low weight for their age. This difference can be attributed to various factors, such as not comparatively analyzing the complexity of the heart diseases presented or the prevalence of comorbidities. Aspects of preoperative clinical management, such as feeding methods, nutritional care, and medication management, were also not described or analyzed.

To categorize the patients in this cohort, the RACHS-1 classification was used [2,11]. Logically, with greater complexity and, therefore, higher mortality risk (higher RACHS-1 score), there is increased inflammation, malnutrition, and, consequently, a greater number and intensity of complications [2,9]. In our population, the higher the RACHS-1 score, the longer the time for chylous effusion to appear. This can be explained by common postoperative clinical management measures and their consequences, such as increased cumulative fluid balance, sustained elevated central venous pressure due to the presence of a venous catheter or hypervolemia, and the occurrence of deep vein thrombosis [2,16,22]. These more clinically ill children typically experience a longer fasting period in the immediate postoperative period and therefore may take longer to manifest chylous effusion. Thus, it is likely that the delay in diagnosing and managing chylous effusion may cause the perpetuation and intensification of the inflammatory state. Therefore, it can be deduced that early diagnosis and therapeutic intervention in cases of chylous effusion is beneficial for the evolution of these patients.

In older and heavier patients, the diagnosis of chylous effusion was made earlier. This is likely due to technical factors, such as the child’s lymphatic circulation volume, which increases proportionally with growth and is therefore more easily detectable [9].

Based on the data obtained here and the analysis of the literature, it is possible to suggest that some of the detected chylous effusions may not result from direct surgical injury, but rather from a persistently severe clinical state, with mechanisms that impair lymphatic drainage, intense inflammation, and progressively worsen the nutritional status, leading to hypoalbuminemia [4,20,22].

When analyzing all these data together, we emphasize that clinical and laboratory monitoring of postoperative cardiac surgery patients is fundamental for the early diagnosis of chylous effusion and timely treatment to prevent clinical deterioration. In addition, some mineral imbalances, such as hypophosphatemia and hypocalcemia, are associated with high mortality and morbidity, due to arrhythmias, muscle weakness, and gastrointestinal disturbances [23].

After all, chylous effusion is known to be a serious comorbidity that increases hospitalization time, intensive care, nutritional therapy, and hospital costs [2,4,22], and is associated with a worsening clinical state in general, where the patient becomes more inflamed and malnourished the longer they remain undiagnosed and untreated [21].

## 5. Study Limitations

This study has some limitations in reaching definitive conclusions. First, our sample size was small for some of the statistical analyses initially proposed. Second, two deaths occurred; as negative outcomes are a significant point of morbidity analysis, it was not possible to establish statistical significance in the correlations.

Additionally, we highlight that the analysis presented here was extracted retrospectively from the HNTT data collection forms and therefore daily clinical progression data, especially related to the therapies used in patient management, were not analyzed. However, it is important to add that the hospital’s HNTT maintains a protocol-based data collection system to facilitate case management and study.

Nonetheless, this sample represents a rare complication in the pediatric population and a specific morbidity, providing relevant discussions and as mentioned earlier, emphasizing the importance of early and rigorous monitoring for timely diagnosis and therapeutic intervention.

Thus, analyzing this population with a larger number of cases, in addition to observing all clinical information and therapies received, may provide greater clarity in understanding chylous effusions in pediatric patients undergoing cardiac surgery.

## 6. Conclusions

Likely because the sample consisted predominantly of inflamed children, the presence of lymphopenia, hypoalbuminemia, and low HDL-cholesterol levels were frequent. There was a high incidence of mineral disturbances, particularly hypocalcemia and hypophosphatemia, which, if left untreated, may lead to unfavorable outcomes.

Although we cannot make formal recommendations based on our retrospective cohort study, we can speculate that active monitoring of inflammation and of the presence of cavity effusion, through non-invasive methods, should be routine in children following cardiac surgery. Therefore, through this monitoring, we can aid in the early diagnosis of chylous effusion and, consequently, timely initiation of treatment.

## Figures and Tables

**Table 1 nutrients-16-03845-t001:** Demography in pediatric patients with chylous effusion ^a^.

Variable	No. (%)
Sex (%)	
Male	13 (56.5)
Female	10 (43.5)
Nutritional Status	
Malnutrition	14 (60.9)
Eutrophic	9 (39.1)
Outcome	
Medical Discharge	19 (82.6)
Death	2 (8.7)
Transferred	2 (8.7)
Rachs	
1	5 (21.7)
2	3 (13.0)
3	11 (47.8)
4	4 (17.4)
Rachs Type	
1	3 (13.0)
2	20 (87.0)
Hyponatremia (%)	4 (17.4)
Hypokalemia (%)	5 (21.7)
Hypocalcemia (%)	10 (43.4)
Hypomagnesemia (%)	1 (13.0)
Hypophosphatemia (%)	6 (26.1)
Lymphopenia (%)	19 (83.0)
Hypoalbuminemia (%)	17 (73.9)

^a^ Values are presented as No. (%). Reference values: hyponatremia (1 month to 1 year: <129 mmol/L; >1 year: <135 mmol/L); hypokalemia (1 month to 1 year: <3.6 mmol/L; >1 year: <3.1 mmol/L); hypocalcemia (<1.15 mmol/L); hypophosphatemia (<4 mg/dL); hypomagnesemia (up to 6 years: <1.2 mEq/L; 6 to 12 years: <1.2 mEq/L); lymphopenia (under 2 years: <3000 × 10^6^/L; over 2 years: <1500 × 10^6^/L); and hypoalbuminemia (<3.5 g/dL).

**Table 2 nutrients-16-03845-t002:** Laboratory tests at chylous effusion diagnosis.

Parameter	Median (p25, p75)
Lymphocytes (n × 10^6^/L)	1910 (1000, 3130)
Sodium (mmol/L)	136 (132, 139)
Potassium (mmol/L)	3.8 (3.4, 4.5)
Magnesium (mEq/L)	1.45 (1.29, 1.78)
Ionized calcium (mmol/L)	1.16 (1.07, 1.2)
Phosphorus (mg/dL)	4.6 (3.7, 5.0)
Albumin (g/dL)	3.2 (2.4, 3.5)
Creatinine (mg/dL)	0.3 (0.25, 0.39)
HDL cholesterol (mg/dL)	24 (17.5, 28.0)
CRP (C-reactive protein) (mg/L)	37.7 (31.2, 66)

**Table 3 nutrients-16-03845-t003:** Correlation between variables and time of chylous effusion onset.

		Time of Chylous Effusion Onset	
Variable	Median (p25, p75)	Rho	*p* Value
Albumin	3.2 (2.4, 3.5)	−0.434	**0.039**
Lymphocytes	1910 (1000, 3130)	−0.169	0.440
HDL	24 (17, 28)	−0.429	0.052
CRP	64 (33.77, 88.75)	0.597	0.068
Triglycerides in effusion	423 (253, 811)	−0.099	0.655
RACHS-1	3 (2, 3)	0.636	**0.001**
Fasting time (days)	7 (4, 9)	−0.046	0.834
Age (months)	6 (3, 11)	−0.485	**0.019**
Weight (kg)	4.8 (4, 7)	−0.550	**0.007**

Rho: Spearman’s rank correlation coefficient. Bold values highlight statistical significance (*p* < 0.05).

## Data Availability

The original contributions presented in the study are included in the article, further inquiries can be directed to the corresponding author.

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
