# Peer review of "Nutritional, Metabolic, and Inflammatory Alterations in Children with Chylous Effusion in the Postoperative Period of Cardiac Surgery: A Descriptive Cohort"

_nutrients, 2024, doi:10.3390/nu16223845_

Round 1
Reviewer 1 Report
Comments and Suggestions for Authors
Dear Corresponding Author, thank you for submitting your article to Nutrients and congratulations on your work.
Brief Summary
Your study addresses a clinically relevant topic, describing metabolic and nutritional alterations in children with chylous effusion after cardiac surgery. The retrospective analysis of 23 cases provides interesting informations about correlations between time of effusion onset and various clinical/biochemical parameters.
General Comments
- The manuscript is well structured and the methodology is generally apropriate for a retrospective observational study. The statistical analysis is also adequate, although some integrations could be useful; the discussion section is well articulated and effectively connects the results to existing literature
Specific Comments
a) Introduction:
- Lines 41-44: It would be useful to provide specific data on the increase in cases in recent years
- Lines 46-48: Consider adding more recent references to support the statement about morbidity/mortality
b) Methods:
- The methodological section could benefit from more details about exclusion criteria because they were not clear to me
- Lines 80-81: Specify how the analysis was handled in case of missing data and why you made this choice
- Sample size calculation is not mentioned or I was unable to find it, please explain better.
c) Results:
- Tables 1,2 and 3: Consider adding cumulative percentages, I think it would be appropriate but it's not a mandatory request; also the characters and dimensions are always completely different from the journal's required template and creates confusion when reading.
- Lines 136-140: The correlation results would deserve a more detailed presentation
d) Discussion:
- The study limitations section is honest but could be expanded and probably included in a dedicated section.
- Consider adding a subsection on practical implications of the results or expand in the discussion, and they could be more specific regarding clinical recomendations
These modifications would further improve the quality and impact of your work. We do not want to request rejection of the work because it is undoubtedly worthy of publication but I invite the authors to review and consider some minor revisions suggested.
I look forward to reading a final version for definitive approval.
Author Response
|
Summary |
|
|
|
Thank you very much for taking the time to review this manuscript. Please find the detailed responses below and the corresponding corrections highlighted in the re-submitted files. Please, find also below our comments on your suggestions and requests.
|
||
|
Point-by-point response to Comments and Suggestions for Authors |
||
|
Comments 1: Lines 41-44: It would be useful to provide specific data on the increase in cases in recent years
Response 1: We agree with the importance of pointing this out, so we decided to include a direct reference for Dr Mery´s article, appointed as our Reference number 2 |
||
|
|
||
|
Comments 2: Lines 46-48: Consider adding more recent references to support the statement about morbidity/mortality
Response 2: Agree. We, accordingly to your suggestion, added a reference to Brandewie et al, 2024, to support the statement about morbidity and mortality |
||
|
|
||
|
Comments 3: The methodological section could benefit from more details about exclusion criteria because they were not clear to me
Response 3: We respectfully inform you that all available cases within the period described were included, therefore, there were no exclusion criteria, as the aim of the study was to describe our cohort, and not to perform a specific analysis.
Comments 4: Lines 80-81: Specify how the analysis was handled in case of missing data and why you made this choice
Response 4: Our Table 1 presents our descriptive whole data. All 23 subjects have these information. We considered HDL and CPR as markers for inflammation, as discussed in the article. On table 2, we described laboratory tests. We could extract the complete data for all the subjects, except 1 patient, that hadn´t neither HDL, nor CPR. Therefore, he was not included in the analysis. As it was not the main objective of our study, we didn´t describe it in the text. If you consider this analysis to be essential, we may add it to the final text.
Comments 5: Sample size calculation is not mentioned or I was unable to find it, please explain better.
Response 5: Again, we thank you for your comment. Nonetheless, chylous effusion is a rare complication of a not so frequent condition. Therefore, we did not perform a sample size calculation. We sought all cases available in our hospital within a convenience period of 11 years. Data collected by our MNTT were regularly available from 2011 till present day.
Comments 6: Tables 1,2 and 3: Consider adding cumulative percentages, I think it would be appropriate but it's not a mandatory request; also the characters and dimensions are always completely different from the journal's required template and creates confusion when reading.
Response 6: Due to the small number of cases enrolled in the categories presented in table 1, we chose not to add cumulative percentages. Related to our tables´ dimensions and characters, they lost our settings in this article´s version, edited by the Journal. We ask you, if possible, to guide us on how to solve this issue.
Comments 7: Lines 136-140: The correlation results would deserve a more detailed presentation
Response 7: |
||
|
We thank you again for pointing it out. We changed the paragraph to include a better description of our results.
Comments 8: The study limitations section is honest but could be expanded and probably included in a dedicated section.
Response 8: We also agreed and transformed the study limitations in a dedicated section
Comments 9: Consider adding a subsection on practical implications of the results or expand in the discussion, and they could be more specific regarding clinical recommendations
Response 9: In regard of your comment, we decided to add the following words: Although we cannot make formal recommendations based on our retrospective cohort study, we can speculate that active monitoring of inflammation and of the presence of cavity effusion, through non-invasive methods, should be routine in children following cardiac surgery. Further studies directed at this hypothesis are necessary. |
||
|
|
||
|
Additional clarifications
Finally, we thank you for all your comments and contributions and we remain available for further explanations and changes that might be necessary. |
||
|
|
||

Reviewer 2 Report
Comments and Suggestions for Authors
This manuscript studies the profile of clinical, nutritional, and metabolic characteristics associated with chylous effusion following pediatric cardiac surgery
Several changes should be done
1. The total number of patients who participated in the study should be included in the abstract.
2. Include epidemiological information, what is the incidence or prevalence of this alteration,
3. Material and methods, include the program used to do the statistics, for example R, IBM SPSS, etc.
4. Discuss in the discussion the small sample size (23 patients) and how affect the validity of your information.
5. Explain if there were missing data .(The manuscript said that 2 patients died) , if yes, how did you deal with them
6. In the discussion the authors discuss nutrient and mineral imbalances. Please discuse also with more detail the potential physiological mechanisms
Author Response
|
Summary |
|
|
|
Thank you very much for taking the time to review this manuscript. Please find the detailed responses below and the corresponding corrections highlighted in the re-submitted files. Please, find also below our comments on your suggestions and requests.
|
||
|
Point-by-point response to Comments and Suggestions for Authors |
||
|
Comments 1: The total number of patients who participated in the study should be included in the abstract.
Response 1: We agree with the importance of pointing this out, so we made this change. |
||
|
|
||
|
Comments 2: Include epidemiological information, what is the incidence or prevalence of this alteration,
Response 2: Agree. We added, in the introduction section, the incidence´s range of chylous effusion in children that undergo cardiac surgery. |
||
|
|
||
|
Comments 3: Material and methods, include the program used to do the statistics, for example R, IBM SPSS, etc.
Response 3: Also agreed. We added this information on the Material and methods section.
Comments 4: Discuss in the discussion the small sample size (23 patients) and how affects the validity of your information.
Response 4: We respectfully accept your observation. We have already commented on this subject in the discussion section, in a separate paragraph, to highlight this point. As we dealt with a rare complication, that occurs in about 5% of patients with a rare disease (cardiac abnormalities), we chose to conduct only a descriptive study.
Comments 5: Explain if there were missing data .(The manuscript said that 2 patients died) , if yes, how did you deal with them
Response 5: We are glad to inform you that the 2 patients that died and the 2 patients that were referred to another hospital were included in the analysis, because the event (death or referral) occurred after the chylous effusion onset. Therefore, these events did not interfered in our analysis.
Comments 6: In the discussion the authors discuss nutrient and mineral imbalances. Please discuss also with more detail the potential physiological mechanisms
Response 6 We thank you for the comment and we agree. Therefore, we added a new paragraph to emphasize these physiological mechanisms.
|
||
|
|
||
|
|
||
|
Additional clarifications
Finally, we thank you for all your comments and contributions and we remain available for further explanations and changes that might be necessary. |
||
